# GRAPHICAL RESIDUAL FLOWS

**Jacobie Mouton**
Computer Science Division
Stellenbosch University
South Africa

**Steve Kroon** *
Computer Science Division, Stellenbosch University AND
National Institute for Theoretical and Computational Sciences
South Africa

## ABSTRACT

Graphical flows add further structure to normalizing flows by encoding non-trivial variable dependencies. Previous graphical flow models have focused primarily on a single flow direction: the normalizing direction for density estimation, or the generative direction for inference. However, to use a single flow to perform tasks in both directions, the model must exhibit stable and efficient flow inversion. This work introduces graphical residual flows, a graphical flow based on invertible residual networks. Our approach to incorporating dependency information in the flow, means that we are able to calculate the Jacobian determinant of these flows exactly. Our experiments confirm that graphical residual flows provide stable and accurate inversion that is also more time-efficient than alternative flows with similar task performance. Furthermore, our model provides performance competitive with other graphical flows for both density estimation and inference tasks.

## 1 INTRODUCTION

Normalizing flows (NFs) (Tabak & Turner, 2013) have proven to be a useful tool in many machine learning tasks, such as density estimation (Papamakarios et al., 2017) and amortized inference (Kingma et al., 2016). These models represent complex probability distributions as bijective transformations of a simple base distribution, while tracking the change in density through the change-of-variables formula. Bayesian networks (BNs) model distributions as a structured product of conditional distributions, allowing practitioners to specify expert knowledge. Wehenkel & Louppe (2020) showed that the modelling assumptions underlying autoregressive transformations used in NFs correspond to specific classes of BNs. Building on this, Wehenkel & Louppe (2021) propose a graphical NF that encodes an arbitrary BN structure via the NF architecture. They only consider the class of flows that enforces a triangular Jacobian matrix. Weilbach et al. (2020) instead extend the continuous NF of Grathwohl et al. (2019) by encoding a BN through the sparsity of the neural network's weight matrices. These graphical flow approaches focus on only one flow direction: either the normalizing direction for density estimation or the generative direction for inference. They do not emphasize stable inversion. NFs are theoretically invertible, but stable inversion is not always guaranteed in practice (Behrmann et al., 2021): if the Lipschitz constant of the inverse flow transformation is too large, numerical errors may be amplified. Another class of flows, known as residual flows (Chen et al., 2019), obtain stable inversion as a byproduct of their Lipschitz constraints, but do not encode domain knowledge about the target distribution's dependency structure.

This work proposes the graphical residual flow (GRF), which encodes domain knowledge from a BN into a residual flow in a manner similar to previous graphical NFs. GRFs capture a predefined dependency structure through masking of the residual blocks' weight matrices. Stable and accurate inversion is achieved by constraining a Lipschitz bound on the flow transformations. The incorporation of dependency information also leads to tractable computation of the exact Jacobian determinant. We compare the GRF to existing approaches on both density estimation and inference tasks. Our experiments confirm that this method yields competitive performance on both synthetic and real-world datasets. Our model exhibits accurate inversion that is also more time-efficient than alternative graphical flows with similar task performance. GRFs are therefore an attractive alternative to existing approaches when a flow is required to perform reliably in both directions.

---

*Correspondence to `kroon@sun.ac.za` .

## 2 GRAPHICAL RESIDUAL FLOW

Consider a residual network $F(\boldsymbol{x}) = (f_T \circ \ldots \circ f_1)(\boldsymbol{x})$, composed of blocks $\boldsymbol{x}^{(t)} := f_t(\boldsymbol{x}^{(t-1)}) = \boldsymbol{x}^{(t-1)} + g_t(\boldsymbol{x}^{(t-1)})$, with $\boldsymbol{x}^{(0)} = \boldsymbol{x}$. $F$ can be viewed as a NF if all of its component transformations $f_t$ are invertible. A sufficient condition for invertibility is that $\mathrm{Lip}(g_t) < 1$, where $\mathrm{Lip}(\cdot)$ denotes the Lipschitz constant of a transformation. Behrmann et al. (2019) construct a *residual flow* by applying spectral normalization to the residual network's weight matrices such that the bound $\mathrm{Lip}(g_t) < 1$ holds for all layers. The graphical structure of a BN can be incorporated into a residual flow by suitably masking the weight matrices of each residual block before applying spectral normalization. Given a BN graph, $\mathcal{G}$, over the components of $\mathbf{x} \in \mathbb{R}^D$, the update to $\mathbf{x}^{(t-1)}$ in block $f_t$ is defined as follows for a residual block with a single hidden layer—it is straightforward to extend this to residual blocks with more hidden layers:

$$\mathbf{x}^{(t)} := \mathbf{x}^{(t-1)} + (W_2 \odot M_2) \cdot h((W_1 \odot M_1) \cdot \mathbf{x}^{(t-1)} + b_1) + b_2 \ . \tag{1}$$

Here $h(\cdot)$ is a nonlinearity with $\mathrm{Lip}(h) \leq 1$, $\odot$ denotes element-wise multiplication, and the $M_i$ are binary masking matrices ensuring that component $j$ of the residual block's output is only a function of the inputs corresponding to $\{\mathrm{x}_j\} \cup \mathrm{Pa}_{\mathcal{G}}(\mathrm{x}_j)$. By composing a number of such blocks, each variable ultimately receives information from its ancestors in the BN via its parents. This is similar to the way information propagates between nodes in message passing algorithms. The masks above are constructed according to a variant of MADE (Germain et al., 2015) for arbitrary graphical structures (see Appendix A.1-2). The parameters of the flow as a whole are trained by maximizing the log likelihood through the change-of-variables formula: $\log p(\mathbf{x}) = \log p_0(F(\mathbf{x})) + \log |\det (J_F(\mathbf{x}))|$, where $\det (J_F(\mathbf{x}))$ denotes the Jacobian determinant of the flow transformation.

Since we are enforcing a DAG dependency structure between the variables, there will exist some permutation of the components of $\mathbf{x}$ for which the correspondingly permuted version of the Jacobian would be lower triangular. We can thus compute $\det(J_F(\mathbf{x}))$ *exactly* as the product of its diagonal terms—which is invariant under such permutations. This is in contrast to standard residual flows, which require approximation of the Jacobian determinant.

**Inversion**   The inverse of this flow does not have an analytical form (Behrmann et al., 2019). Instead, each block can be inverted numerically using either a Newton-like fixed-point method (Song et al., 2019) or the Banach fixed-point iteration method (Behrmann et al., 2019). Since the convergence rate of the latter is dependent on the Lipschitz constant, we expect the former to perform better when larger Lipschitz bounds for the residual blocks ($\approx 0.99$) are used. To compute $\mathbf{x} = f_t^{-1}(\mathbf{y})$, the Newton-like fixed-point method applies the following update until convergence:

$$\mathbf{x}^{(n)} = \mathbf{x}^{(n-1)} - \alpha(\mathrm{diag}(J_{f_t}(\mathbf{x}^{(n-1)})))^{-1}[f_t(\mathbf{x}^{(n-1)}) - \mathbf{y}], \tag{2}$$

using the initialization $\mathbf{x}^{(0)} = \mathbf{y}$ and letting $0 < \alpha < 2$, which ensures local convergence (Song et al., 2019). Note that this convergence is not necessarily to the correct $\mathbf{x}$: we implicitly rely on the assumption that it is unlikely inversion will fail using the initialization $\mathbf{x}^{(0)} = \mathbf{y}$.

**Inference**   In the case where latent variables $\mathbf{z}$ are present, one typically only has access to the forward BN that models the generating process for an observation $\mathbf{x}$. That is, the BN generally encodes the following factorization of the joint: $p(\mathbf{x}, \mathbf{z}) = p(\mathbf{x}|\mathbf{z})p(\mathbf{z})$. To perform inference, we first invert the BN structure using the faithful inversion algorithm of Webb et al. (2018). This allows us to construct a graphical residual flow where the latent variables are conditioned on the observations: $\mathbf{z}^{(t)} := \mathbf{z}^{(t-1)} + (W_2 \odot M_2) \cdot h((W_1 \odot M_1) \cdot (\mathbf{z}^{(t-1)} \oplus \mathbf{x}) + b_1) + b_2$, where $\oplus$ denotes concatenation. When used to train a variational inference artifact $q$, the objective is to maximize the evidence lower bound (ELBO) (Kingma & Welling, 2014), $\mathbb{E}_{\mathbf{z} \sim q}[\log p(\mathbf{x}, \mathbf{z}) - \log q(\mathbf{z}|\mathbf{x})]$.

**Activation Function**   Since the loss functions above contain the derivatives of the residual block activation functions through the Jacobian term, the gradients used for training will depend on the second derivative of the activation function. It is thus desirable to use smooth *non-monotonic* activation functions that adhere to the imposed Lipschitz bounds (such as LipSwish Chen et al. (2019)) to avoid gradient saturation. In our model, we use an activation we call "LipMish"— $\mathrm{LipMish}(x) = (x/1.088) \tanh(\zeta (\zeta(\beta) \cdot x))$. This is a scaled version of the non-monotonic Mish activation function (Misra, 2020) satisfying $\mathrm{Lip}(\mathrm{LipMish}) \leq 1$ for all $\beta$. The coefficient of $x$ is parameterized to be strictly positive by first passing $\beta$ through a softplus function, $\zeta(\cdot)$.

Table 1: Density estimation and inference performance. Each entry indicates the average log-likelihood (for density estimation) or ELBO (for inference) on the test set over five runs, with the standard deviation given in the subscript. Bold indicates the best results in each model size category.

| | Density Estimation (LL) | | | Inference (ELBO) | |
|---|---|---|---|---|---|
| | Arithmetic Circuit | Tree | Protein | Arithmetic Circuit | Tree |
| GNF-A$_S$ | $-1.30_{\pm 0.02}$ | $-9.35_{\pm 0.00}$ | $-5.47_{\pm 0.20}$ | $-5.40_{\pm 0.35}$ | $-2.33_{\pm 0.00}$ |
| GNF-M$_S$ | $\mathbf{-1.29}_{\pm \mathbf{0.11}}$ | $\mathbf{-8.65}_{\pm \mathbf{0.00}}$ | $4.51_{\pm 0.15}$ | $-4.69_{\pm 0.20}$ | $\mathbf{-1.71}_{\pm \mathbf{0.00}}$ |
| SCCNF$_S$ | $-1.79_{\pm 0.41}$ | $-8.77_{\pm 0.03}$ | $3.17_{\pm 0.34}$ | $\mathbf{-4.55}_{\pm \mathbf{0.21}}$ | $-1.82_{\pm 0.06}$ |
| GRF$_S$ | $-1.46_{\pm 0.04}$ | $-8.67_{\pm 0.01}$ | $\mathbf{4.78}_{\pm \mathbf{0.10}}$ | $-4.78_{\pm 0.38}$ | $-1.74_{\pm 0.01}$ |
| GNF-A$_L$ | $-1.27_{\pm 0.07}$ | $-9.30_{\pm 0.00}$ | $-1.56_{\pm 0.40}$ | $-4.88_{\pm 0.26}$ | $-2.45_{\pm 0.00}$ |
| GNF-M$_L$ | $\mathbf{-1.07}_{\pm \mathbf{0.05}}$ | $\mathbf{-8.56}_{\pm \mathbf{0.00}}$ | $6.44_{\pm 2.00}$ | $\mathbf{-4.02}_{\pm \mathbf{0.16}}$ | $-1.72_{\pm 0.00}$ |
| SCCNF$_L$ | $-1.17_{\pm 0.33}$ | $-8.62_{\pm 0.05}$ | $7.18_{\pm 0.15}$ | $-4.25_{\pm 0.40}$ | $-1.75_{\pm 0.02}$ |
| GRF$_L$ | $-1.10_{\pm 0.01}$ | $-8.58_{\pm 0.00}$ | $\mathbf{7.54}_{\pm \mathbf{0.05}}$ | $-4.42_{\pm 0.12}$ | $\mathbf{-1.70}_{\pm \mathbf{0.00}}$ |

## 3 EXPERIMENTS

We compare our proposed GRF to two existing approaches—the graphical normalizing flow of Wehenkel & Louppe (2021), for which we consider both affine and monotonic transformations (denoted by GNF-A and GNF-M, respectively) and the structured conditional continuous normalizing flow (SCCNF) presented in Weilbach et al. (2020). We use the arithmetic circuit dataset (Weilbach et al., 2020), an adaptation of Wehenkel & Louppe (2021)'s tree dataset, and a real-world human proteins dataset (Sachs et al., 2005) (see Appendix B.1). To provide more informative comparisons between the flows, we train two models per task for each approach. The first is a smaller model with a maximum capacity of 5000 trainable parameters, which will be denoted by a subscript S, e.g., GRF$_S$. We also train a larger model with a maximum capacity of 15000 parameters, denoted by the subscript L. All flows were trained using the Adam optimizer for 200 epochs with an initial learning rate of $0.01$ and a batch size of 100. The learning rate was decreased by a factor of 10 each time no improvement in the loss was observed for 10 consecutive epochs, until a minimum of $10^{-6}$ was reached. The training and test sets of the synthetic datasets consisted of 10 000 and 5000 instances, respectively. The protein dataset was divided into 5000 training instances and 1466 test instances. For further information on the flow architectures and experimental setup, see Appendix B.2 and B.3. Our experiments consider the relative performance of the flows with respect to tasks in both the generative and normalizing directions, as well as the efficiency and accuracy of flow inversion.

### 3.1 DENSITY ESTIMATION AND INFERENCE PERFORMANCE

Table 1 provides the log-likelihood (LL) and ELBO achieved by each model on the test sets for density estimation and inference tasks, respectively. For density estimation we assume all variables to be observed and for inference a subset is taken to be latent. GRF achieves results similar to GNF-M and SCCNF, with a close second-best score for most datasets, and the best scores on the protein dataset (density estimation) and for the large model on the tree dataset (inference). GNF-A, with its reliance on affine transformations, is unable to provide matching performance.

### 3.2 INVERSION

We first confirm the computational advantage of using the Newton-like inversion procedure of Equation 2 to invert the flow steps of a GRF, rather than the Banach fixed-point iteration method. Figure 1 illustrates that the convergence rate of the latter is heavily dependent on the Lipschitz bound on the residual block—here controlled by the hyperparameter $c$.

We next explore the inversion stability of GRFs compared to graphical models with similar task performance, namely GNF-M and SCCNF. GNF-M and GRF were inverted using the procedure of Equation 2. We optimize the hyperparameters $\alpha$ and the number of iterations per flow step ($N$) with a grid search. The stopping criterion for the number of iterations $N$

Table 2: Comparison of the inversion performance for the different flow models on 100 test data points from the protein dataset. Bold indicates best results in each model size category. Ranges indicate different optimal settings for $N$ and $\alpha$ for different data points.

| Flow | Small Models | | | | Large Models | | | |
|---|---|---|---|---|---|---|---|---|
| | Converged within 50 steps | $N$ | $\alpha$ | Inversion time (ms) | Converged within 50 steps | $N$ | $\alpha$ | Inversion time (ms) |
| GNF-M | 87 | 9–46 | 0.3–1.0 | 117.28 | 89 | 5–49 | 0.3–1.0 | 371.38 |
| SCCNF | 96 | – | – | 73.45 | 94 | – | – | 277.05 |
| GRF | **100** | 5–8 | 1.0 | **63.69** | **100** | 5–8 | 1.0 | **157.57** |

was a reconstruction error of less than $10^{-4}$. To better illustrate potential inversion instability, we perform this optimization on a per data point basis for 100 data points from the protein dataset (whereas in practice one would typically invert the sample as a single batch). SCCNF was inverted by executing the integration in the opposite direction. We record the number of data points for which the desired reconstruction error is achieved, and where $N \leq 50$ in the case of GRF and GNF-M. We also measure the time it takes to invert the flow for the entire batch using the settings that allow the most data points to have the desired reconstruction error. The results are given in Table 2. One of the main paradigms for enforcing global stability is using Lipschitz-constrained flow transformations (Behrmann et al., 2021). In the case of the GRF, this stability is automatically achieved as a byproduct of the flow design, and we see that GRF shows excellent inversion accuracy. GNF-M, depending on the architecture and learned weights, has either potentially very large Lipschitz bounds, or has no global Lipschitz bounds at all. This helps to explain its poorer inversion results. While SCCNFs have global Lipschitz bounds, these are not controlled during training and numerical instability can thus occur. For further information see Appendix A.3.

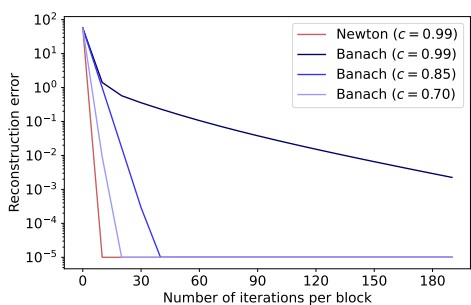

Figure 1: Equation 2 requires far fewer iterations per block to invert $GRF_S$ than the Banach fixed-point approach. The plot shows the average reconstruction error for 100 samples from the arithmetic circuit dataset.

## 4  CONCLUSION

We proposed the graphical residual flow as an alternative to existing NFs that incorporate dependency information from a predefined BN into their architecture. Including domain knowledge about the dependencies between variables leads to much sparser weight matrices in the residual blocks, but still allows sufficient information to propagate in order to accurately model the data distribution. Although existing methods like GNF-M (Wehenkel & Louppe, 2021) and SCCNF (Weilbach et al., 2020) provide very good modelling capability in a given flow direction, inverting these flows for other tasks can be unstable and computationally expensive. In our experiments, GNF-A provided a compact model with fast and accurate inversion. It does not, however, achieve very good modelling results on more complex datasets. The graphical residual flow provides an alternative trade-off between modelling capability and inversion stability and efficiency. It provides primary task performance comparable to the best obtained by the other models, while providing the most stable and quickest inversion performance: flows can be inverted with few fixed-point iterations, and without taking special care to control the step size. Unlike other models considered, the GRF's blocks are globally bi-Lipschitz, with desired Lipschitz bounds enforced during training, and this may explain the stable inversion observed. The GRF is thus a suitable option to use in situations where some dependency structure is assumed to exist and where the flow may be required to perform reliably in both directions. The GRF may also have further application in problems such as BN structure learning, which can be explored in future work.

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

## A  Normalizing Flows with Graphical Structures

### A.1  Graphical Residual Flow Illustration

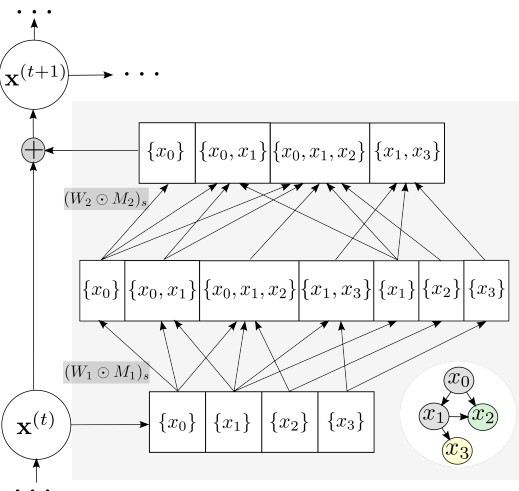

Figure A.1: An illustration of the update $\mathbf{x}^{(t+1)} := \mathbf{x}^{(t)} + (W_2 \odot M_2)_s \cdot h((W_1 \odot M_1)_s \cdot \mathbf{x}^{(t)} + b_1) + b_2$ applied to $\mathbf{x}$ at each flow step $t$ of a graphical residual flow where each residual block has a single hidden layer. Subscript $s$ indicates spectral normalization after masking. The edges removed by the masks $M_1$ and $M_2$ are not shown; the remaining edges encode the graphical structure of the given BN. To avoid cluttering the diagram, the bias terms are omitted. The assignment of sets to nodes in the residual block's neural network is discussed in Section A.2.

### A.2  Enforcing a Graphical Structure through Masking

Suppose a joint distribution factorizes as

$$p(\mathbf{x}) = \prod_{i=1}^{D} p(x_i | \text{Pa}_{x_i}^{\mathcal{G}}) \tag{3}$$

for DAG $\mathcal{G}$. $\text{Pa}_{x_i}^{\mathcal{G}}$ denotes the parents of $x_i$ in $\mathcal{G}$. For a given neural network that takes $\mathbf{x}$ as input, the goal is to have the output units associated with $x_i$ be computed from only those input units associated with $x_i$ and its parents. This means that there should be no computational paths between an input and an output unit if there is no direct dependency between the associated variables in $\mathcal{G}$. This can be achieved by applying a masking matrix to the weights of each neural network layer (which can be of arbitrary width) such that at least one weight on any such computational path is set to zero.

We follow a similar approach to MADE (Germain et al., 2015), which constructs masks for an implicit fully-connected BN. We begin by assigning a specific subset of variables to each unit in the neural network. Specifically, each input unit is assigned a unit set containing its corresponding input variable: $\{x_i\}$. Each output unit is assigned a set consisting of its associated variable and that variable's parents in the BN: $\{x_i\} \cup \text{Pa}_{x_i}^{\mathcal{G}}$. Lastly, each hidden unit is randomly assigned one of the following sets: $\{x_i\}$ or $\{x_i\} \cup \text{Pa}_{x_i}^{\mathcal{G}}$ where $i$ can be any of $1, \ldots, D$.[1]

A correct mask can then be constructed by ensuring that it zeroes out any weight between two neural network units if the set assigned to the unit in the next layer is not a superset of the set assigned to the unit in the previous layer. This has the implication that any path from input to output for any variable has a single associated set switch from $\{x_i\}$ to $\{x_i\} \cup \text{Pa}_{x_i}^{\mathcal{G}}$.

---

[1]To prevent situations where there are no valid paths from an input to the corresponding output, we also require at least one unit in each hidden layer associated with $\{x_i\}$.

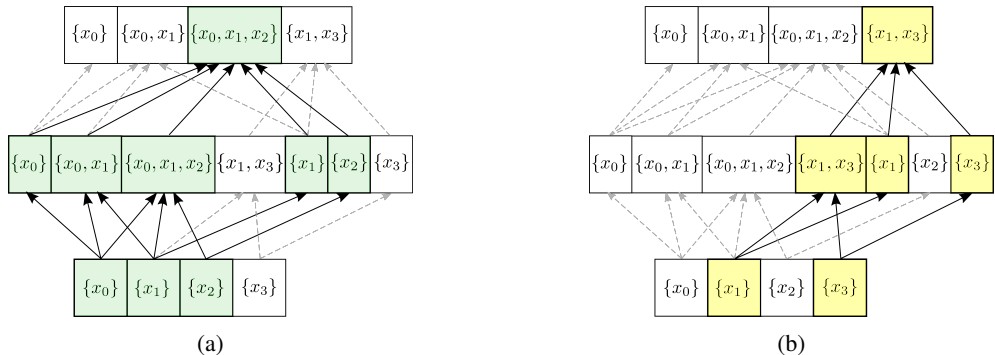

Figure A.2: An example application of the above masking scheme for the graphical residual flow illustrated in Figure A.1. The masks are constructed by assigning each output node the set consisting of the associated variable and its parents in the BN. Each hidden node is assigned a set consisting of either a single variable or a variable and its parents. An edge is retained only if the set in the next layer is a superset of the set in the previous layer. Thus, the updates to variables $x_0$, $x_1$, $x_2$ (a) and $x_3$ (b) are only conditioned on their respective parents.

## A.3 INVERTIBILITY OF GRAPHICAL NORMALIZING FLOWS IN PRACTICE

The Lipschitz constants of the forward and inverse transformation of an invertible neural network quantifies its worst-case stability. Bounds on these values play an important role in understanding and mitigating possible exploding inverses.

**GNF-A**   No global bounds can be placed on the Lipschitz constant of a GNF with affine normalizers. This complicates the task of ensuring stable inversion in all scenarios. Behrmann et al. (2021) provide the following simple illustration of this. Assume $\mathbf{x}$ consists of two variables, $x_0$ and $x_1$, and consider the transformation $F_1(\mathbf{x}) = x_1 \cdot s(x_0)$ as the component of a normalizer corresponding to $x_1$. This is affine in $x_1$, with some arbitrarily complex non-constant function $s$ arising from the conditioner. Then

$$\frac{\partial F_1(\mathbf{x})}{\partial x_0} = x_1 \frac{\partial s(x_0)}{\partial x_0}. \tag{4}$$

If we take into account both the identity,

$$\text{Lip}(F) = \sup_{\mathbf{x} \in \mathbb{R}^D} ||J_F(\mathbf{x})||_2, \tag{5}$$

and that the derivative is unbounded since $x_1$ may grow arbitrarily large, Behrmann et al. (2021) argue that the unbounded Jacobian can induce an unbounded spectral norm and thus no global Lipschitz bound can be obtained.

**GNF-M**   We can employ a similar illustration to investigate the Lipschitz bounds of a GNF with monotonic normalizers. Again assume $\mathbf{x}$ consists of two variables, $x_0$ and $x_1$, where $x_1$ depends on $x_0$ in the corresponding BN. The transformation applied to $x_1$ is then given by $F_1(\mathbf{x}) = \int_0^{x_1} f(t, h_1(x_0)) dt + \beta(h_1(x_0))$ (Wehenkel & Louppe, 2021). We take its partial derivative, using Leibniz's integral rule and the chain rule:

$$\frac{\partial F_1(\mathbf{x})}{\partial x_0} = \frac{\partial}{\partial x_0} \int_0^{x_1} f(t, h_1(x_0)) \, dt + \frac{\partial \beta(h_1(x_0))}{\partial x_0} \tag{6}$$

$$= \int_0^{x_1} \frac{\partial f(t, h_1(x_0))}{\partial h_1(x_0)} \frac{\partial h_1(x_0)}{\partial x_0} \, dt + \frac{\partial \beta(h_1(x_0))}{\partial x_0} \ .$$

The integrand here is the product of the derivatives of two neural networks with respect to their inputs. For general networks, it is not necessarily the case that this integral will be bounded, since the integrand's shape will depend on not only the chosen activation functions, but also the weights obtained during training. Thus, either the architecture of the flow must be adapted to ensure that this integral remains bounded for any $x_1 > 0$, or other techniques must be used to improve local stability, as discussed in Behrmann et al. (2021).

**SCCNF** Given that the flow is defined by a neural ODE, $\frac{d\mathbf{x}(t)}{dt} = F(\mathbf{x}(t), t)$, where $t \in [0, 1]$, we have that the Lipschitz constant for both the forward and inverse transformation are upper bounded by $e^{\text{Lip}(F) \cdot t}$ (Behrmann et al., 2021).

## B DATASETS AND EXPERIMENTS

### B.1 BAYESIAN NETWORK DATASETS

**Arithmetic Circuit** The arithmetic circuit BN follows the same structure as the generative network used by Weilbach et al. (2020) and Wehenkel & Louppe (2021). For density estimation tasks, all variables are observed. For amortized inference tasks, variables $z_0$ to $z_5$ are latent, while $x_0$ and $x_1$ are observed. This distribution consists of heavy-tailed densities which are linked through non-linear dependencies:

$$
\begin{aligned}
z_0 &\sim \text{Laplace}(5, 1) \\
z_1 &\sim \text{Laplace}(-2, 1) \\
z_2 &\sim \mathcal{N}(\tanh(z_0 + z_1 - 2.8), 0.1) \\
z_3 &\sim \mathcal{N}(z_0 \times z_1, 0.1) \\
z_4 &\sim \mathcal{N}(7, 2) \\
z_5 &\sim \mathcal{N}(\tanh(z_3 + z_4), 0.1) \\
x_0 &\sim \mathcal{N}(z_3, 0.1) \\
x_1 &\sim \mathcal{N}(z_5, 0.1).
\end{aligned}
$$

**Tree** This is another synthetic dataset. It is adapted from the model given in Wehenkel & Louppe (2021), to obtain a known model for which the joint distribution can be computed (as needed for the inference tasks). Instead of using the circles 2D dataset from Grathwohl et al. (2019), as used in Wehenkel & Louppe (2021), the first two variables are sampled from a 2D Gaussian mixture model, $\text{GMM}_2$, which consists of two equally weighted components with means at $(1, 1)$ and $(-1, -1)$. As in Wehenkel & Louppe (2021), the second pair of variables is sampled from a GMM with 8 equally weighted components with means at $(0, 1.5)$, $(1, 1)$, $(1.5, 0)$, $(1, -1)$, $(0, -1.5)$, $(-1, -1)$, $(-1.5, 0)$ and $(-1, 1)$.

$$
\begin{aligned}
z_0, z_1 &\sim \text{GMM}_2 \\
z_2, z_3 &\sim \text{GMM}_8 \\
z_4 &\sim \mathcal{N}(\max(z_0, z_1), 1) \\
z_5 &\sim \mathcal{N}(\min(z_2, z_3), 1) \\
x_0 &\sim \mathcal{N}\left(\frac{1}{2}(\sin(z_4 + z_5) + \cos(z_4 + z_5)), 1\right)
\end{aligned}
$$

**Protein** This dataset consists of 11 observed variables containing information about multiple phosphorylated human proteins (Sachs et al., 2005). The BN structure encodes the cellular signaling network that is believed to exist between these proteins. The dataset is divided into 5000 and 1466 training and test instances, respectively.

### B.2 FLOW ARCHITECTURES

We train two models per task for each of the approaches. The first is a smaller model with a maximum capacity of 5000 trainable parameters, denoted by a subscript S. The second, larger model has a maximum capacity of 15000 parameters, denoted by the subscript L. Table B.1 details the flow architectures used for each of the datasets. These architectures were obtained by performing a grid search over flow depth and neural network width, maximizing the number of weights while adhering to the model size budget. Each residual block and conditioner neural network of the GRF and GNF models had a single hidden layer, respectively. The same architectures were used for both density estimation and inference. The integral neural network of all monotonic flows consisted of one hidden layer of size 100, with ELU activation functions on both the hidden and final layer. The

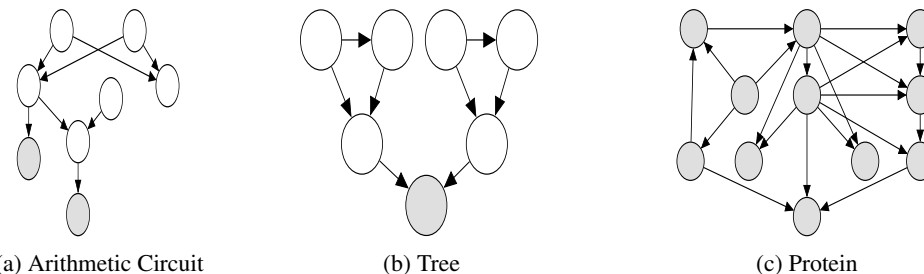

(a) Arithmetic Circuit        (b) Tree        (c) Protein

Figure B.1: BN graphs associated with each dataset. White nodes indicate latent variables, whereas grey nodes are observed. When used for density estimation, all nodes are taken to be observed.

| | Flow | Number of weights | Nr of flow steps | Hidden layer size | Activation function |
|---|---|---|---|---|---|
| **Arithmetic Circuit** | GNF-A$_S$ | 4464 | 4 | 200 | ReLU |
| | GNF-M$_S$ | 4592 | 2 | 50 | ReLU |
| | SCCNF$_S$ | 1120 | 2 | 150 | tanh |
| | GRF$_S$ | 4320 | 8 | 125 | LipMish |
| | GNF-A$_L$ | 14994 | 9 | 300 | ReLU |
| | GNF-M$_L$ | 14680 | 4 | 125 | ReLU |
| | SCCNF$_L$ | 9316 | 4 | 150 | tanh |
| | GRF$_L$ | 14569 | 17 | 200 | LipMish |
| **Tree** | GNF-A$_S$ | 4260 | 4 | 200 | ReLU |
| | GNF-M$_S$ | 4486 | 2 | 50 | ReLU |
| | SCCNF$_S$ | 1160 | 2 | 150 | tanh |
| | GRF$_S$ | 4616 | 8 | 125 | LipMish |
| | GNF-A$_L$ | 14301 | 9 | 300 | ReLU |
| | GNF-M$_L$ | 14132 | 4 | 125 | ReLU |
| | SCCNF$_L$ | 10412 | 4 | 150 | tanh |
| | GRF$_L$ | 14490 | 21 | 150 | LipMish |
| **Protein** | GNF-A$_S$ | 4604 | 4 | 150 | ReLU |
| | GNF-M$_S$ | 4831 | 1 | 125 | ReLU |
| | SCCNF$_S$ | 1274 | 2 | 150 | Tanh |
| | GRF$_S$ | 4779 | 9 | 100 | LipMish |
| | GNF-A$_L$ | 13815 | 9 | 200 | ReLU |
| | GNF-M$_L$ | 14493 | 3 | 125 | ReLU |
| | SCCNF$_L$ | 8362 | 4 | 150 | Tanh |
| | GRF$_L$ | 14586 | 22 | 125 | LipMish |

Table B.1: Flow architectures.

hyperparameter $c$, constraining the spectral norm in the graphical residual flow, was set 0.99 in all cases.

All flows were trained using the Adam optimizer for 200 epochs with an initial learning rate of 0.01 and training batches of size 100. The learning rate was decreased by a factor of 10 each time no improvement in the loss was observed for 10 consecutive epochs, until a minimum of $10^{-6}$ was reached.

## B.3 INVERSION

We use the inversion algorithm of Song et al. (2019) as given in Algorithm 1. The hyperparameters $\alpha$ and $N$ are optimized using a grid search to find the setting that produces the fastest inversion with a reconstruction error less than $10^{-4}$. Values for $\alpha$ in the range $0.1 \times t$, for $t = 1, \ldots, 19$, were considered.

---

**Algorithm 1** Fixed-point iteration to compute $\mathbf{x} = f_t^{-1}(\mathbf{y})$

---

**Require:** $N$, $0 < \alpha < 2$
  Initialize $\mathbf{x}_0 \leftarrow \mathbf{y}$
  **for** $n \leftarrow 1$ to $N$ **do**
    Compute $f_t(\mathbf{x}_{n-1})$
    Compute $\mathrm{diag}(J_{f_t}(\mathbf{x}_{n-1}))$
    $\mathbf{x}_n \leftarrow \mathbf{x}_{n-1} - \alpha(\mathrm{diag}(J_{f_t}(\mathbf{x}_{n-1})))^{-1}[f_t(\mathbf{x}_{n-1}) - \mathbf{y}]$
  **end for**
  **return** $\mathbf{x}_N$

---

Figures B.2 to B.5 show the reconstruction error (log-scale) for varying numbers of iterations used at each step while inverting the flow, and for different values of $\alpha$. In Figure B.2, we see that both GNF-A and GRF require only a few iterations and one can simply use the setting $\alpha = 1$ for all datasets. More care needs to be taken when choosing a value for $\alpha$ to invert GNF-M. While Figure B.3 looks promising for GNF-M, Figures B.4 and B.5 show GNF-M's varying inversion performance for different data points from the arithmetic circuit and protein test sets. Clearly, one cannot simply set $\alpha = 1$ for these models, as some instances may require a much smaller step size to ensure stable inversion, which subsequently requires more iterations to converge. The stable inversion of all test instances by the small and large GNF-M models may be due to the simplicity of the tree distribution and the absence of any outlier or noisy data points.

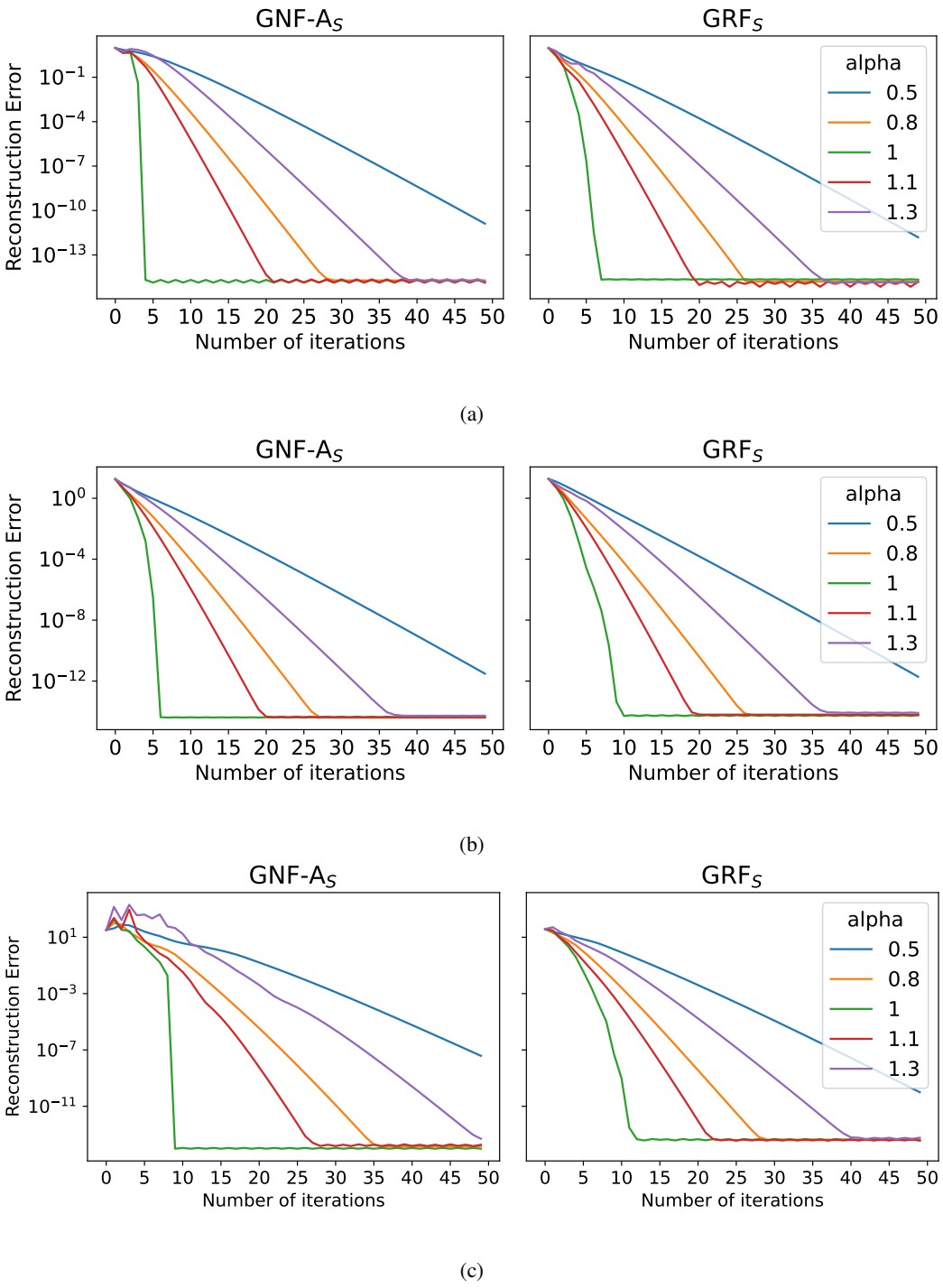

Figure B.2: Reconstruction error for different values of $\alpha$ and $N$ when inverting GNF-A$_S$ and GRF$_S$ for the (a) arithmetic circuit, (b) tree and (c) protein datasets. The larger models showed similar results.

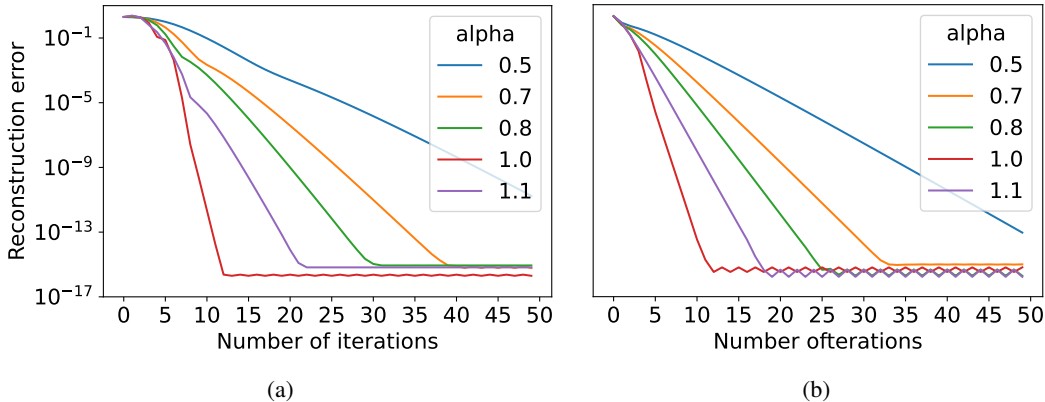

Figure B.3: Finding optimal $\alpha$ and $N$ for (a) GNF-M$_S$ and (b) GNF-M$_L$ for the tree dataset. This is the only dataset for which the GNF with monotonic normalizers showed stable inversion.

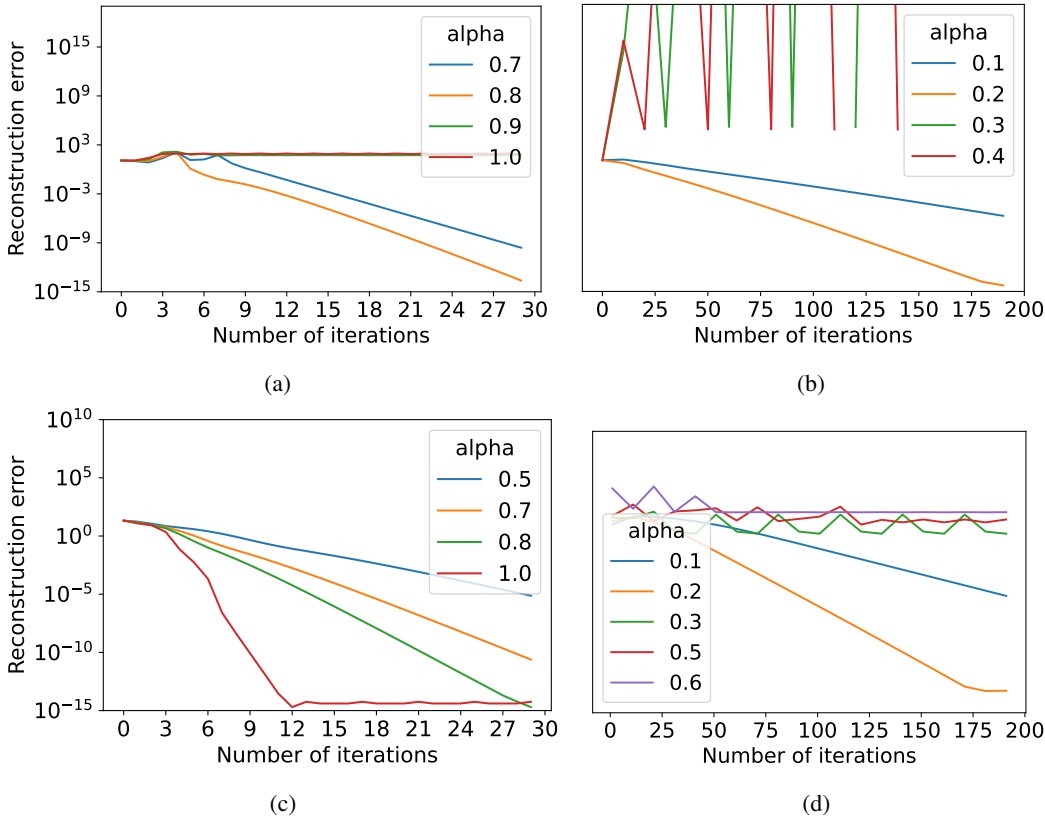

Figure B.4: Reconstruction error for different values of $\alpha$ and $N$ when inverting GNF-M$_S$ (top row) and GNF-M$_L$ (bottom row) for the arithmetic circuit dataset. The left column shows a case for which the flow inversion converges within a reasonable number of steps. The right column illustrates that the flow can exhibit poor convergence on certain data points. Note the change in scale on the horizontal axis.

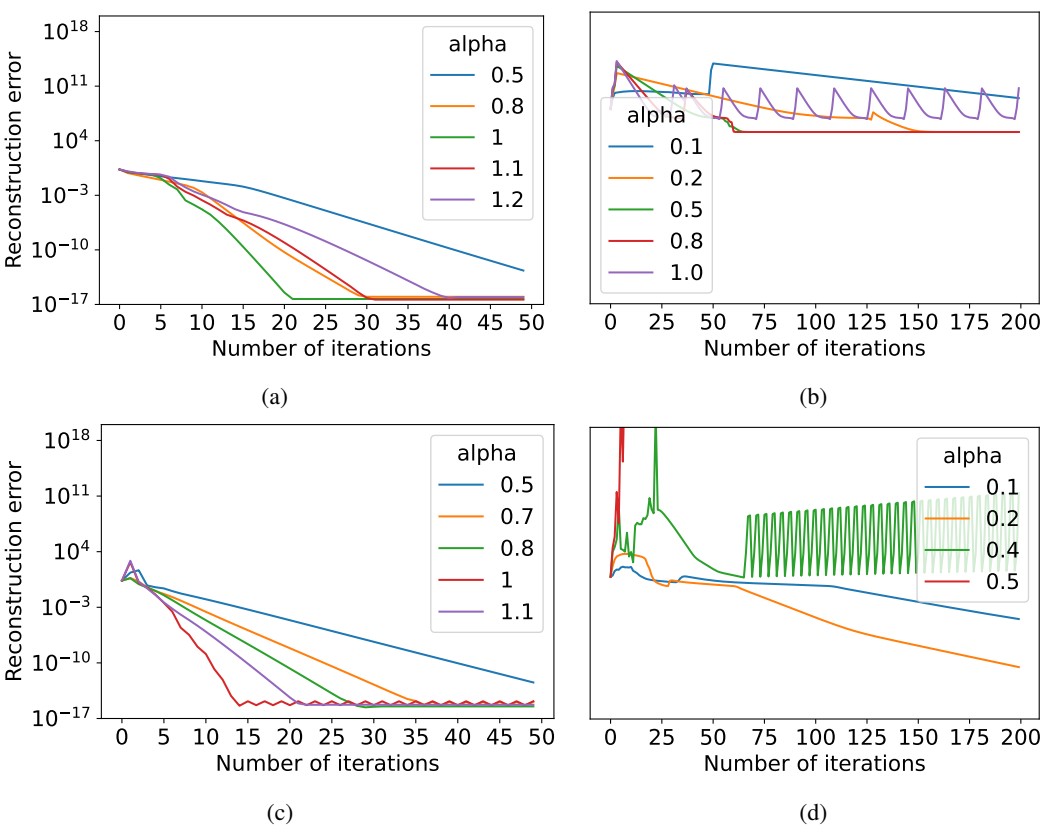

Figure B.5: Reconstruction error for different values of $\alpha$ and $N$ when inverting GNF-M$_S$ (top row) and GNF-M$_L$ (bottom row) for the protein dataset. The left column shows a case for which the flow inversion converges within a reasonable number of steps. The right column illustrates that the flow can exhibit poor or no convergence on certain data points. Note the change in scale on the horizontal axis.