# OpenReview forum: "Graphical Residual Flows"
_ICLR.cc/2022/Workshop/DGM4HSD — ICLR 2022 DGM4HSD workshop Poster_

### Official Review · Reviewer_aHEt · 2022-03-12
**Graphical Residual Flows Review**

**Rating:** 7
**Confidence:** 5

**Review:**

This paper proposes Graphical Residual Flows, which is a modification of the popular Residual Flow architecture with masking over the weight matrices that enable the incorporation of BN or DAG. Furthermore, the papers demonstrate that GRFs ---in contrast to vanilla Residual Flows don't need an approximation to the $\text{det} | J_F(x) |$. The paper also proposes that a more stable inversion can be achieved using the Newton-like fixed-point method proposed by Song et al. (2019) as opposed to the Banach Fixed-point method employed in the original paper. The experiments demonstrate that empirically inversion is both more stable and faster in convergence in terms of the number of steps required. Finally, the authors apply GRFs to a variety of datasets for both density estimation and maximizing the ELBO and show comparable performance to other flow based approaches. Overall, this is a nice paper with an interesting idea. There are few minor areas of improvement in the paper outlined below that could help the overall readability as well as improve the technical quality.

- The introduction can be made more precise, as the majority of the discussion is focused on Residual Flows but is presented as it is universally true for all NFs. For example, NFs don't necessarily have to be theoretically invertible. There have been past work where the flow map is only surjective---which induces a slightly different change of variable formula (e.g. Principal Manifold Flows).
- An illustrative figure that improves in conceptualizing the idea.
- Synthetic toy datasets which require explicit control using a BN should be considered to demonstrate the effectiveness of GRFs over vanilla RFs or other flow methods to incorporate BNs.
- "Since we are enforcing a DAG dependency structure ... would be lower triangular." This is an interesting insight but I had a hard time understanding this, or why this is guaranteed to be true. I encourage the authors to explain this point in a bit more detail, perhaps even providing a more technical justification (e.g. Proposition, theorem, etc ...)

---

### Official Review · Reviewer_4vPP · 2022-03-21
**Missing some clarification how the work advances the current state of understanding**

**Rating:** 5
**Confidence:** 5

**Review:**

**Summary:**

The submission proposes a modification of Residual Normalizing Flows [Behrmann 2019] with a structured Jacobian which allows exact and efficient computation of the Jacobian determinant, as well as, inverting the flow efficiently using Newton's method. It further introduces a new activation function called *LipMish* which promises non-saturation of gradients during the optimization. Preliminary experiments on density estimation tasks show comparable results to similar methods. A second experiment show-cases the efficiency improvement from using Newton's method over the fix-point interaction introduced by [Behrmann 2019].

**Significance:**

The idea of using structured matrices to represent the residual update is interesting and convincing. The explanation is technically sound and seems correct. Experiments show comparable results to similar SOTA methods.

I have a couple of concerns that I would like to have addressed to recommend this submission for the workshop:

Major concerns:

I am afraid the authors should put some thought into how to discriminate their work from similar prior work. At the end of the first paragraph of Sec. 2. they explain that the structured Jacobian will be triangular up to permutations. This implies the architecture has a close resemblance to other works using triangular Jacobians, e.g., [Gopal 2020] or [DeCao 2018] (see Sec. 3.1. at the end) just to name two. As the masks are fixed, am I right that the presented work corresponds to AR flows with fixed intermediary permutations? It would be very helpful if the authors could make a clear discussion here.

My second line of concerns regards the Newton method:
1. when reading the manuscript I have been a bit confused about the presentation of the Newton inversion of [Song 2019] here. Is this is not merely applying their result to this particular architecture choice? If so, why is there a need for additional experimental validation?
2. the authors correctly state that Newton's method can show more rapid convergence compared to other methods like fixpoint iterations or bisection. However, Newton's method applied to non-convex problems can also show non-stationary/cyclic behavior (e.g. think of inverting a perfectly symmetric sigmoid) so that the initial condition is critical for convergence. Assuming that the flows are possibly very complex non-convex transformations, I do not see this aspect discussed here.

Minor concern:

I believe the related work section could need some improvement. E.g., residual flows with structured Jacobians allowing for the exact computation of the determinant are also given, e.g., by planar flows [Rezende 2015] or Sylvester normalizing flows [Van Den Berg 2018]. It would be helpful for the reader to see things in context.

**Writing and presentation:**
The paper is clearly written and easy to read for researchers in the domain. An illustratory figure comparing the presented architecture to alternative architectures (black-box residual flows, GNFs, AR flows) could be helpful to process the core innovation of the idea quickly.

**Bottom-line:***
Due to the concerns above, I am currently not sure whether I think this submission is ready for the workshop. I am primarily concerned about a lacking discussion of prior work and putting things in context. If the authors address my concerns above, I am happy to accept it.

---


[Bermann 2019] Behrmann, Jens et al. “Invertible Residual Networks.” ICML (2019).

[Gopal 2020] Gopal, Achintya. "Quasi-Autoregressive Residual (QuAR) Flows." arXiv preprint arXiv:2009.07419 (2020).

[De Cao 2018] De Cao, Nicola, Wilker Aziz, and Ivan Titov. "Block neural autoregressive flow." Uncertainty in artificial intelligence. PMLR, 2020.

[Song 2019] Yang Song, Chenlin Meng, and Stefano Ermon. MintNet: Building invertible neural networks with
masked convolutions. In Advances in Neural Information Processing Systems, pp. 11004–11014.
Curran Associates, Inc., 2019.

[Rezende 2015] Rezende, Danilo, and Shakir Mohamed. "Variational inference with normalizing flows." International conference on machine learning. PMLR, 2015.

[Van Den Berg 2018] Van Den Berg, Rianne, et al. "Sylvester normalizing flows for variational inference." 34th Conference on Uncertainty in Artificial Intelligence 2018, UAI 2018. Association For Uncertainty in Artificial Intelligence (AUAI), 2018.

---

### Official Review · Reviewer_DBne · 2022-03-23
**Great contribution**

**Rating:** 8
**Confidence:** 4

**Review:**

The paper proposes a variant of a _graphical_ normalizing flow, which allows encoding a known dependency structure into the flow, or, in some cases, _learning_ this dependency structure. Prior work explores such methods in the context of autoregressive/coupling flows and continuous flows. The authors consider another important class of flows: _residual_ flows. The experiments demonstrate that the proposed graphical flow provides a more efficient inverse (likely due to the constrained Lipschitz constant), while retaining likelihoods competitive with prior methods.

I think the paper contains plenty of technical novelty, is written well, evaluates the proposed method carefully, and is upfront about its trade-offs and the context in which it can be useful. The paper is rather dense, which is understandable given the tight page limit, but I wonder if some of the experiments (e.g. inference experiments with corresponding text in Section 2) could be moved to the appendix, giving the rest of the paper more breathing space.

For completeness (probably in the future full paper), I'd like to see a few words on how the $M_i$ matrices are constructed for the various datasets used. I agree with the authors that _learning_ the matrices $M_i$ (via their relaxation) is an interesting future direction. In the full paper I would also like to see more ablations for the choice of the activation function and the inverse method. Is the "LipMish" indeed an improvement over the previously used "LipSwish"? Can the Newton-like fixed-point method be used for prior graphical flows, and would it similarly improve the inverse efficiency?

In summary, I think this paper will be an excellent contribution to the workshop, encouraging discussion around graphical normalizing flows, and presenting a method with a different set of trade-offs. I recommend accepting the paper.

---

### Decision · Program_Chairs · 2022-03-27

Accept (Poster)